# *Globorotalia truncatulinoides* in the Mediterranean Basin during the Middle–Late Holocene: Bio-Chronological and Oceanographic Indicator

**Giulia Margaritelli** [1,*], **Fabrizio Lirer** [2], **Katrin Schroeder** [3], **Angela Cloke-Hayes** [4], **Antonio Caruso** [5], **Lucilla Capotondi** [6], **Teresa Broggy** [4], **Isabel Cacho** [7] and **Francisco J. Sierro** [8]

1   Istituto di Ricerca per la Protezione Idrogeologica (IRPI), CNR, Via Madonna Alta 126, 06128 Perugia, Italy
2   Dipartimento di Scienze della Terra, Sapienza University of Rome, Piazzale A. Moro 5, 00185 Roma, Italy; fabrizio.lirer@uniroma1.it
3   Istituto di Scienze Marine (ISMAR), CNR, Arsenale, Tesa 104, Castello 2737/F, 30122 Venezia, Italy; katrin.schroeder@ismar.cnr.it
4   Department of Geography, Mary Immaculate College, University of Limerick, V94 VN26 Limerick, Ireland; angela.hayes@mic.ul.ie (A.C.-H.); teresa.broggy@mic.ul.ie (T.B.)
5   Dipartimento di e Tecnologie Biologiche Chimiche e Farmaceutiche della Terra e del Mare (STEBICEF), Università degli Studi di Palermo, Via Archirafi, 18, 90123 Palermo, Italy; antonio.caruso@unipa.it
6   Istituto di Scienze Marine (ISMAR), CNR, Via Gobetti 101, 40129 Bologna, Italy; lucilla.capotondi@bo.ismar.cnr.it
7   GRC Geociències Marines, Departamento de Dinàmica de la Terra i de l'Oceà, Facultat de Ciències de la Terra, Universitat de Barcelona, 08028 Barcelona, Spain; icacho@ub.edu
8   Departamento de Geología, Universidad de Salamanca, 37008 Salamanca, Spain; sierro@usal.es
*   Correspondence: giulia.margaritelli@irpi.cnr.it

**Abstract:** The planktonic foraminiferal species *Globorotalia truncatulinoides* is widely used as a biostratigraphic proxy for the Quaternary in the Mediterranean region. High-resolution quantitative studies performed on sediment cores collected in the central and western Mediterranean Sea evidence a significant abundance of *G. truncatulinoides* during the Middle Holocene. The robust chronological frame allows us to date this bio-event to 4.8–4.4 ka Before Present (BP), very close to the base of the Meghalayan stage (4.2 ka BP). As a consequence, we propose that *G. truncatulinoides* can be considered a potential marker for the Middle–Late Holocene chronological subdivision. *G. truncatulinoides* is a deep-dwelling planktonic foraminifer and their distributional pattern in the central and western Mediterranean Sea provides a tool to monitor the onset of the regional deep vertical mixing of the water column. During the Holocene, the significant increase in the abundance of this species is in phase with the end of African Humid Period, which marks the transition from a more humid climate to the present-day semi-arid climate.

**Keywords:** *Globorotalia truncatulinoides*; Meghalayan stage; 4.2 event; vertical mixing; Mediterranean Sea

## 1. Introduction

The identification and characterization of bio-events are fundamental in recognizing the stratigraphic units contributing to the reconstruction of Earth history. Specifically, different species of planktonic foraminifera have been extensively used to determine the biostratigraphical chronology of deep-sea sediments. Among these, *Globorotalia truncatulinoides*, recently renamed *Truncorotalia truncatulinoides* in the phylogenetic review by Aze et al. (2011) [1], is one of the most common planktonic species used for the chronological characterization of Mediterranean stratigraphical sequences during the Quaternary [2].

The Holocene Epoch represents the uppermost chronostratigraphic unit within the geological time scale and covers the time interval from 11.7 ka Before Present (BP) until the present day [3]. The Holocene is characterized by significant climate variability [4–10] that, in recent centuries, has also been influenced by anthropogenic activities [11–13].

In 2018, the Holocene Epoch was formally subdivided into three stages/ages: the Greenlandian (starting at 11.7 ka BP), Northgrippian (8.2 ka BP) and the Meghalayan (4.2 ka BP). The two former stages/ages are supported by Global Boundary Stratotype Section and Points (GSSPs) from Greenland ice cores, whereas the Meghalayan GSSP was obtained from a speleothem located in the Mawmluh Cave in northeastern India [14]. The Meghalayan age begins on 4.2 ka BP, corresponding to the so-called abrupt-change event characterized by dry climatic conditions in many parts of the world [15] from North America, through the Middle East to China, and from Africa, parts of South America, and Antarctica [16–18].

However, this event is only documented in a few marine and continental records in the Mediterranean region [5,19–26]. In marine sediments from the western Mediterranean Sea, the base of the Meghalayan stage is identifiable by oxygen-stable isotopes and SST records [27], but it is necessary to find possible bio-events that could be used to approximate the base of this chronostratigraphic unit.

Recent studies from the central and south Tyrrhenian Sea and the Sicily Channel evidenced a significant increase in the abundance of the planktonic foraminifer left-coiled (l.c.) *G. truncatulinoides* at roughly 4.4 ka BP [5,28,29]. These results suggest this bio-event can be considered as a potential candidate to approximate the base of the Meghalayan stage in the Mediterranean region.

Nowadays, *G. truncatulinoides* is indicative of deep vertical mixing during the winter season [5,30], warranting further exploration on the potential significance of this species as indicative of new oceanographic conditions close to the base of Meghalayan stage in the Mediterranean Sea.

This work aims to (1) evaluate the applicability of the planktonic foraminifer l.c. *G. truncatulinoides* as a potential tool for the chronological subdivision of the Middle–Late Holocene Epoch and (2) explore the distribution of l.c. *G. truncatulinoides* as proxy for late Holocene paleoenvironmental conditions.

For this research, we combine new and previous literature data obtained from marine sediment cores collected across the central–western Mediterranean Sea sub-basins.

## 2. Oceanography of the Study Area: Present-Day Conditions

The semi-enclosed and elongated Mediterranean Sea is characterized by an anti-estuarine circulation, meaning that it is forced by a net evaporation that occurs over its surface, which induces a marked salinity difference with the Atlantic Ocean [31]. The surface water coming from the Atlantic enters the Gibraltar Strait, and spreads throughout the entire Mediterranean Sea. The Atlantic Water (AW) occupies the upper part (100–200 m) of the water column, but the depth range changes regionally. The net evaporation and the mixing with adjacent water masses contribute to the progressive modification of the AW salinity, which increases from ~36.5‰ at Gibraltar to approximately 38.0–38.5‰ in the western Mediterranean and >39‰ in the easternmost part of the basin [32,33] (Figure 1). As the modification of the surface water continues, a number of saltier Mediterranean Waters are formed in different areas of the basin due to intense air–sea interactions. Between the Sardinian and Sicily Channels, the path of the AW splits into two branches [34]: one enters the Tyrrhenian Sea, while the remaining branch flows into the eastern Mediterranean as the Atlantic Tunisian Current and the Atlantic Ionian Stream (Figure 1). The Tyrrhenian Sea is characterized by anticyclonic and cyclonic eddies [35] and pronounced oligotrophy [36]. Surface and intermediate waters exit the Tyrrhenian Sea through both the southern opening and the relatively shallow Corsica Channel (Figure 1). The resultant Eastern Corsica Current flows along the Ligurian coast, where it is joined by the Western Corsica Current and becomes the Northern Current (NC). The path of the NC is observed along the entire northern boundary of the Western Mediterranean Sea, passing offshore the Gulf of Lion and towards the Balearic Sea [37] (Figure 1).

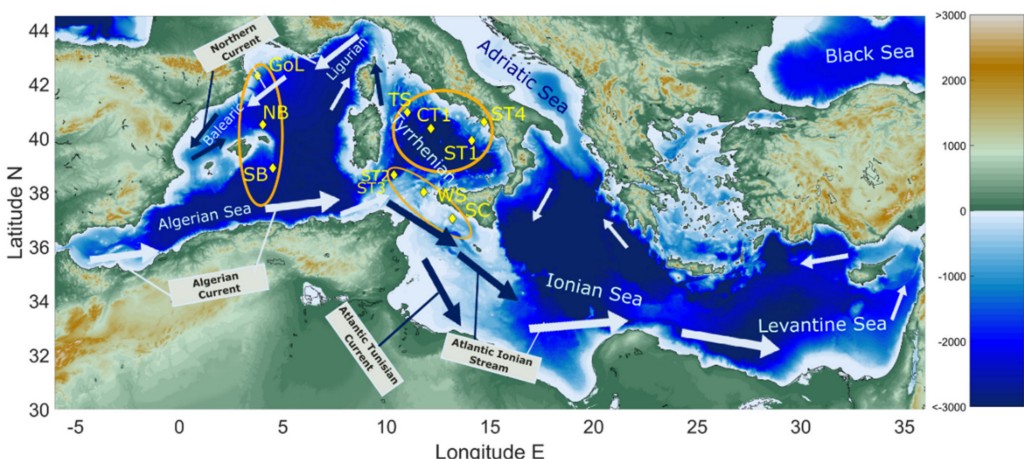

**Figure 1.** Geographical distribution of *G. truncatulinoides* abundance (%) in core top samples of the Mediterranean Sea [38]; MARGO database from Kucera et al. (2005) [39] was modified in this work with additional core top samples from Nextdata Project. The orange circles that divide the study area into three sectors are used for the oceanographic discussions in the text.

During winter, the northwestern part of the basin (i.e., Gulf of Lion) is subjected to intense cooling and evaporation, instigating episodes of vertical convection that may reach the seafloor at depths >2500 m in specific winters, e.g., [40]. In general, this area is characterized by a much deeper mixed layer than other areas of the Mediterranean Sea. Neither the area between the Sardinia Channel and the Sicily Channel nor the Tyrrhenian Sea have these characteristics, even though the mixed layer depth (MLD) shows a clear seasonal cycle.

### 3. Material and Methods

This study was based on 11 marine sediment cores collected in the western part of the Mediterranean Sea (Figure 1) covering the geographical area where the present-day presence of l.c. *G. truncatulinoides* is well documented [41,42] (Figure 2). The main information for each study site is summarized in Table 1; however, the foraminiferal assemblage of ten marine sediment cores of this study were from papers already published (Table 1), while only one (MD99-2343) was reported in this paper for the first time (Table 1). The MD99-2343 core comes from the northern part of the Balearic Sea [43] at 2391 m of water depth. The micropaleontological content of this core is abundant and well-preserved and the age model is improved from that originally published by Frigola et al. (2007) [43], with nine new 14C AMS dates by Català et al., 2019 [27].

A robust chronology of each core already exists: M40/4-82-2SL [44], MD99-2343 [27], ODP 975B [44], CET1 [45], C33 [46], C08 [47], ODP 974B [44], M40/4 80 SL [44], C90 [29], ND11 [30], and NDT6_2016 [10].

All samples were washed using a 63 μm size sieve; then, quantitative planktonic foraminiferal analysis was carried out on splits containing at least 300 specimens. Unfortunately, the different authors did not carry out analyses on the same sediment size fraction: the analyses performed on core C90 were based on the size fraction >90 μm, counts in cores M40/4-82-2 SL, ODP 975B, OPD 974B and M40/4 80 SL were performed on >150 μm, whereas in the remaining records the size fraction was >125 μm. However, this does not invalidate the results, as the census counts performed in the >125 μm and >150 μm fractions reflect a realistic spectrum of the assemblages in the Mediterranean Sea [48].

The left- and right-coiling forms of *G. truncatulinoides* were identified using the conventional method: the test shows right coiling if, when viewed from above with the dorsal side up, the chambers are added in a clockwise direction. In this study, we only considered the left-coiling specimens of *G. truncatulinoides* as it is the most represented in all the sites

in terms of relative abundance [41,42]. The results are reported as percentages of the total faunal assemblage.

For this study, the MLD values of Houpert et al. (2015a) [49] were used. For more details about the data and the methods used, see Houpert et al. (2015b) [50].

**Table 1.** Study sites, depths (meter water depth) and coordinates.

| Core | Area | Water Depth (m) | Coordinates | References |
|------|------|-----------------|-------------|------------|
| M40/4-82-2SL | Gulf of Lion (GoL) | 1079 | 42°18.51′ N, 03°46.40′ E | Broggy PhD thesis |
| MD99-2343 | North Balearic (NB) | 2391 | 40°29.84′ N, 04°01.69′ E | This work (Català et al., 2019 age model) |
| ODP975B | South Balearic (SB) | 2415 | 38°53.78′ N, 04°30.59′ E | Broggy PhD thesis |
| CET1 | South Tyrrhenian Sea (ST) | 2088 | 39°54.69′ N, 14°06.65′ E | Morabito et al., 2014 |
| C33 | South Tyrrhenian Sea (ST) | 2368 | 38°39.48′ N, 10°20.98′ E | Di Stefano et al., 2015 |
| C08 | South Tyrrhenian Sea (ST) | 2370 | 38°38.53′ N, 10°21.55′ E | Budillon et al., 2009 |
| ODP 974B | Central Tyrrhenian Sea (CT) | 3453 | 40° 21.36′ N, 12°08.51′ E | Broggy PhD thesis |
| M40/4 80 SL | Tyrrhenian Sea (TS) | 1881 | 40°57.31′ N, 11°00.22′ E | Broggy PhD thesis |
| C90 | South Tyrrhenian Sea (TC) | 103 | 40°35.76′ N, 14°42.48′ E | Lirer et al., 2013 |
| ND11 | Sicily Channel (SC) | 475 | 37°01′ N, 13°10′ E | Margaritelli et al., 2020 |
| NDT6 | West Sicily (WS) | 1066 | 38°0′ N, 11°47′ E | Trias-Navarro et al., 2021 |

## 4. *G. truncatulinoides*: Habitat and Ecology

Several studies describe *G. truncatulinoides* as a deep-dwelling planktonic foraminifer characterized by a complex life cycle, which involves substantial vertical migration in the water column related to its reproduction [51–57]. The reproduction of this species occurs in late winter at depths where vertical water mixing is required for the migration of juveniles to surface waters [56,58,59]. *G. truncatulinoides* continues its life cycle by migrating down through the water column [60] at ~350 m, reaching cooler waters below the thermocline [56,60–63].

*G. truncatulinoides* has been characterized by five genetically different types (types 1–4) [64] and type 5 [65]. Types 1–4 were identified from genetic data obtained from plankton tows in the Atlantic Ocean and the Mediterranean Sea [64]. The initial temporal events relating to the cladogenesis of *G. truncatulinoides* types 1–4 was established by de Vargas et al. (2001) [64], identifying a differentiation between warm and cold morphotypes at ~300 ka. Renaud and Schmidt (2003) [66] indicated two warm morphotypes (types 1 and 2) at ~170 ka and two cold morphotypes (types 3 and 4) at ~120 ka. Type 5 is present in the northwest Pacific Ocean, only with the dextral-coiling variant [65]. Type 2 and type 5 are the only genetic types that contain the right-coiling variant. Type 5 has, to date, only been identified in oligotrophic subtropical areas in the central water of the North-West Pacific Ocean [65]. While all five genetic types of *G. truncatulinoides* are present in the southern hemisphere, type 2 is the only morphotype that exists in the Northern Hemisphere and in the Mediterranean Sea [64,65,67].

The left- and right-coiling chamber arrangement of *G. truncatulinoides* has been considered to be indicative of different water masses in terms of temperature, salinity, e.g., [52], and depth [68]. At present, a specific study on the coiling direction of *G. truncatulinoides* in the Mediterranean and its possible connection with changes in environmental/oceanographic conditions does not exist. However, in the western Mediterranean, the right-coiled specimens during the Holocene occur only after the chronological interval of Sapropel layer S1 deposition [29,69] and disappear thereafter. It may be suggested that the winter mixing is in favor of the reproductive strategy of the left-coiled form [40,70]. The plankton tow [40,71] and sediment-trap [72] data obtained in the Mediterranean Sea indicate a high abundance of *G. truncatulinoides* during winter and low during sum-

mers [30]. Further, the maximum abundances of *G. truncatulinoides* occur from December to April [73]. The distribution of this species is concentrated in the central–western part of the Mediterranean Sea (Figure 2), in areas of intense water column mixing during the winter months [40]. In contrast, the species is absent from the majority of the eastern basin [30], probably as a result of the ultra-oligotrophy of the easternmost part of the Mediterranean and the inability to survive compared to other regions during winter and spring [74].

A study based on sediment traps and surface sediments highlighted that winter cooling and convective overturning are the primary factors controlling the ecological niche of *G. truncatulinoides* off the northwest African coast [63]. During winter, the surface waters are characterized by a chlorophyll maximum reflecting a phytoplankton bloom, which corresponds with maximum shell abundances [63]. In contrast, during spring and summer, when the seasonal thermocline is re-established, surface waters are depleted of nutrients and the smaller populations of *G. truncatulinoides* are associated with the deep chlorophyll maximum below the mixed layer [63]. This relationship between the seasonal stability of the water column (mixed or stratified) and the associated nutrient enrichment, and the peak abundances of *G. truncatulinoides* has also been observed in the North Atlantic Ocean [54,75], the Bermuda Sea [54], the Western North Atlantic [76], and the Caribbean Sea [52,77].

The recent Mediterranean presence of *G. truncatulinoides* is well documented in core top data from Kallel et al. (1997) [38] and in the MARGO dataset [39] (Figure 2). In particular, a high relative abundance of this species is concentrated in the area from the Menorca basin to the Sicily Channel, while not being present in the Adriatic Sea, and shows low abundances in the Alboran Sea and scattered areas of the Ionian Sea and in the Eastern Mediterranean [30]. Studies on planktonic foraminiferal assemblages over the last four millennia do not document the presence of this species in the Ionian Sea [78], the Adriatic Sea [79], the Aegean Sea [80–83], and the Levantine Sea [74,84].

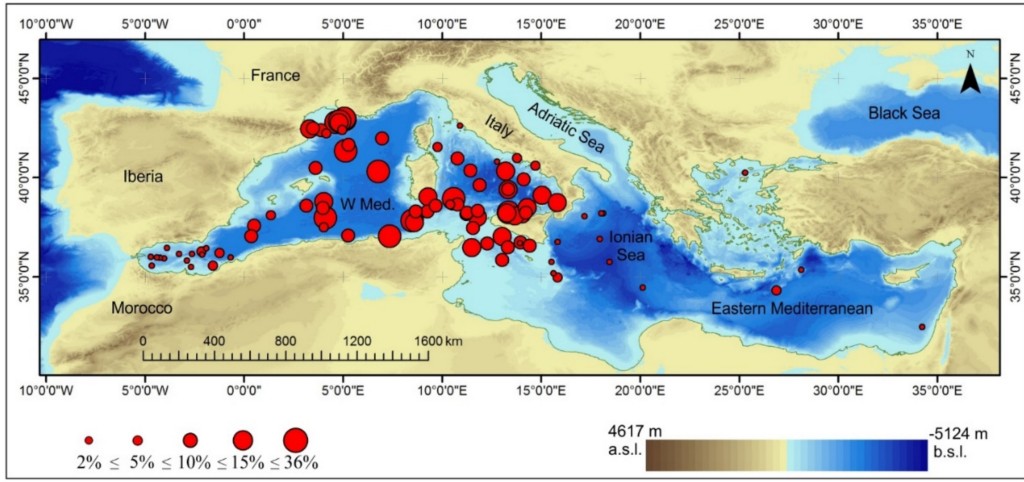

**Figure 2.** Map of the Mediterranean Sea with the sampling points of the study cores (yellow diamonds). GoL = Gulf of Lion; NB = North Balearic; SB = South Balearic; ST = southern Tyrrhenian (1, 2, 3, 4); CT = central Tyrrhenian (1, 2); TS = Tyrrhenian Sea; SC = Sicily Channel; WS = west Sicily (see Table 1).

## 5. Chronological Distribution of *Globorotalia truncatulinoides*

*G. truncatulinoides* originated 2.82 Ma in the Southwest Pacific [85–88]. The species later appeared in the Atlantic Ocean between 2.544 and 2.525 Ma [89], before finally colonizing other ocean basins ~2.0 Ma ago [8,86,89]. More recently, *G. truncatulinoides* has become adapted to colder environments in the Southern Ocean, colonizing subpolar waters in two successive phases of expansion at 300 and 200 kyr [90,91]. According to Cita and Gartner (1973) [92], *G. truncatulinoides* occurred in the Mediterranean Sea much later than in the major ocean basins. Biostratigraphic studies in the Mediterranean Sea have shown that

this species appeared ~0.1 Ma below the Olduvai subchron at ca. 1.81 Ma [2] and became common only ~0.7 Ma later [2,93]. Successively, Di Stefano et al. (1993) [94] and Caruso (2004) [95] adjusted its first occurrence (FO) in MIS 77 with an age of 2.07 and 2.00 Ma, respectively. Moreover, its stratigraphic distribution in younger sediments is rare up to 0.934 Ma [8] (Lirer et al., 2019). This may be due to the mechanism through which *G. truncatulinoides* enters marginal basins (such as the Mediterranean and Caribbean Seas) through shallow and narrow passages, e.g., [77], in relation to the Atlantic circulation controlled by climatic cooling. In fact, *G. truncatulinoides* is able to spread through the open ocean easier because the water column is deeper, unlike at shallow passages, where the column stratification is not ideal for this species.

During the Late Quaternary (the last 140 ka), short peaks of right-coiling *G. truncatulinoides* were documented in the western Mediterranean Sea (Menorca regions) during the isotopic sub-stages 1.01, 2.01, 3.2, 3.3, 5.31, 5.32, 5.4, 6, and 6.4 [96]. However, not all these events are documented in the Levantine basin of the Mediterranean [96,97]. In the SW Pacific, *G. truncatulinoides* has also been extensively used in a new Quaternary biostratigraphic scheme, which adds weight to its use as a marker in the Mediterranean [98].

Later, l.c. *G. truncatulinoides* characterizes the climate phases of the Bølling/Allerød (B/A), the interval between the end of Younger Dryas (YD) and the onset of Sapropel S1 [29,46,47,69,78,79,99–101].

During the last six millennia, l.c. *G. truncatulinoides* has been documented only in the central–western basin: (i) Sicily Channel [69,102,103]; (ii) Tyrrhenian Sea [5,29,45,47,104–107]; and (iii) Balearic Sea [6]. During the last 500 years, l.c. *G. truncatulinoides* has shown a significant increase in abundance during the Maunder Minimum (MM) in the central and western Mediterranean Sea; this time interval is characterized by an atmospheric blocking event [30], which induced an intense deep vertical mixing phenomenon during the winter season, enhancing productivity in the mixed layer. In this context, the ideal ecological conditions for *G. truncatulinoides* proliferation can be produced, suggesting that this species can be considered as an excellent bioindicator of surface water mixing and nutrient availability in the central and western Mediterranean Sea [30].

Conversely, it was absent in the Adriatic, Ionian, and, in general, in the eastern Mediterranean Sea. This marked difference in the geographical distribution of l.c. *G. truncatulinoides* confirms the onset of the modern-day hydrographic conditions in the western Mediterranean Sea, strongly characterized by the development of deep vertical mixing (Figure 2).

## 6. Results and Discussion

### 6.1. Globorotalia truncatulinoides: A Bio-Chronological Indicator

Bio-events can hardly be assumed to be globally synchronous, because the stratigraphic and geographic distribution of species is modulated by ecological preferences exhibited by each taxon and controlled by oceanic circulation, often resulting in earlier or delayed events in certain geographic areas.

In the central–western Mediterranean Sea, several studies have highlighted that l.c. *G. truncatulinoides* temporarily disappears just before the base of Sapropel S1 [99,108].

This species shows a progressive re-occurrence in 5–4.5 ka BP up to the present day (Figure 3). Even if the study sites used for the correlation are characterized by different age models, both in terms of resolution and method, l.c. *G. truncatulinoides* distributional patterns are the same over time in terms of relative abundances across the whole central–western Mediterranean Sea (Figure 3). The relative abundance of the *G. truncatulinodes* never reaches values in excess of 12–16%, excluding core C90 from Salerno Gulf where this taxon reaches its highest values (25–30%) (Figure 3). However, the detected discrepancy is probably due to the different size fraction used for quantitative analyses at this site (>90 micron) [12,29]. In all the investigated sediment cores, the increase in the abundance of l.c. *G. truncatulinoides* occurs in the Middle–Late Holocene transition at the base of the Meghalayan stage (Figure 3), in agreement with the results reported in the Tyrrhenian Sea [5,29].

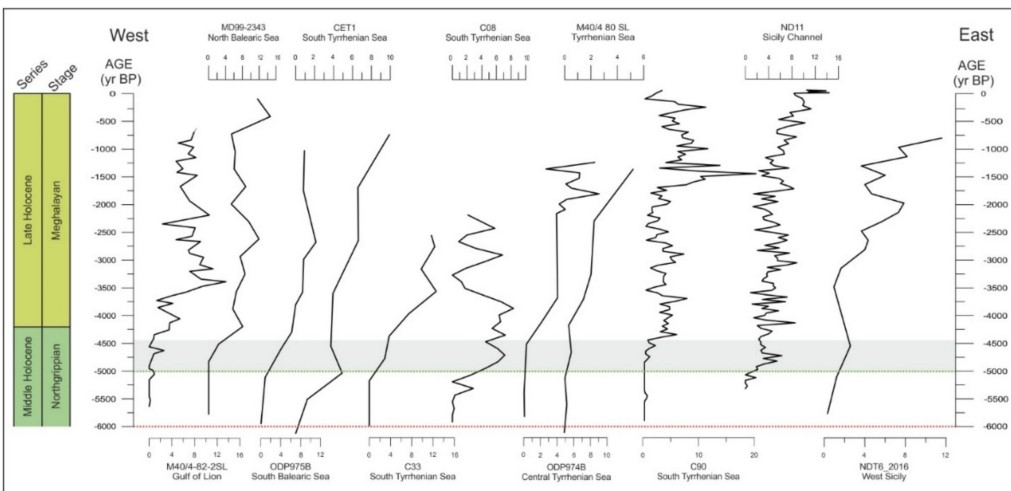

**Figure 3.** Chronological comparisons of *G. truncatulinoides* (left-coiled) abundance percentages (black line) between the marine records: M40/4-82-2SL [44, MD99-2343 [27], ODP975B [44], CET1 [45] (Morabito et al. 2014), C33 [94], C08 [47], ODP 974B [44], M40/4 80 SL [44], C5 [5], C90 [29], ND11 [30], and NDT6_2016 [10]. The gray bar represents the phase in which *G. truncatulinoides* begins to increase in percentage. The red dashed line corresponds to the end of the Sapropel S1 and the green dashed line corresponds to the end of the African Humid Period.

In the Mediterranean Sea, the Middle–Late Holocene transition is characterized by significant paleoclimatic and paleoceanographic changes corresponding to the end of Sapropel S1 layer (c.a. 6 ka BP) [79,108] and to the end of the African Humid climatic period (c.a. 5 ka BP) [109]. These latter Mediterranean changes influenced the distribution pattern of l.c. *G. truncatulinoides* in the central–western Mediterranean Sea.

At ~5.2 ka BP in the Gulf of Lion (core M40/4-82-2 SL), l.c. *G. truncatulinoides* records an increasing trend in relative abundance, culminating in peak frequencies at 3.5 ka BP and remaining consistent up to the present day (Figure 3). Despite the lower resolution, it is possible to observe the same pattern in the Balearic Sea (M99 and ODP 975), starting at 5–4.5 ka BP (Figure 3). Quantitative data detected in the sediments from the southern Tyrrhenian Sea (cores CET1, C33 and C08) evidence an increase in l.c. *G. truncatulinoides* at 5.5 ka BP (about 500 years earlier). At the moment, we do not have an explanation for this. In fact, at 5.5 ka BP, it is possible to observe higher abundances of the species and a continued increasing trend towards the present day (Figure 3). The records of the central Tyrrhenian Sea (cores ODP 974, M40/4 SL) and Salerno Gulf (core C90) align with the trend of the western Mediterranean with an increase in l.c. *G. truncatulinoides*, starting from 5/4.5 ka BP. Lirer et al. (2013) [29] reported the strong increase in l.c. *G. truncatulinoides* abundance in the Salerno Gulf dated at c.a. 4.571 ka BP as a new bio-event useful for western Mediterranean correlation.

Moving to the Sicily Channel (cores ND11, NDT6), this bio-event is chronologically confirmed even in deeper sites starting at c.a. 5 ka BP.

Considering our data, we can assume that, in the central–western Mediterranean Sea during the Middle–Late Holocene, the oceanographic conditions were characterized by enhanced vertical mixing during winter with a strong advection of nutrients from the nutrient-rich deeper layers. This increases the productivity levels in the mixed layer [30], which marks the transition from a more humid climate to the present-day semi-arid climate that is well documented by the distribution pattern of l.c. *G. truncatulinoides*.

### 6.2. Globorotalia truncatulinoides: A Palaeoceanographic Tool

The oceans have a seasonal pattern of stratification, where there is a surface well-mixed layer, a layer where the temperature and other properties change rapidly with depth (the thermocline), and a more uniform deep layer (Figure 4a). The variability of the ML

has a key influence on the physics, chemistry, and biology of the upper ocean, making it one of the influential features for Earth's climate [110]. The base of the ML is generally defined as the region where conditions start to change rapidly with depth (at tens of meters below sea level), and the main driver of the ML deepening is wind, which homogenizes the temperature and salinity of the ML. On the other hand, the warming (or cooling) of the sea surface during the summer (or winter) contributes to the decrease (or increase) in the depth of the ML. During summer, or in warmer regions, such as the eastern Mediterranean Sea (with respect to the western Mediterranean), the warm water remains at the surface and the water column becomes stably stratified; the ML is very shallow, as is the thermocline. On the other hand, during winter, or in cooler regions, such as the western Mediterranean Sea (with respect to the eastern Mediterranean), the surface waters are cooled; this increases its density and makes it sink to its equilibrium depth. In the western basin, the combined action of intense cooling and wind-induced mixing is the cause of a deeper ML. In addition to the horizontal gradient, the depth of the ML base changes on a seasonal basis, increasing during late autumn/winter (from late October to early March) [111] and decreasing again in spring (from April).

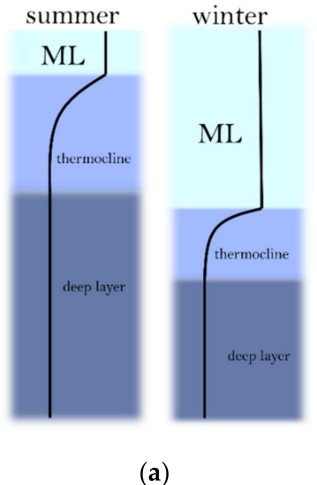

(**a**)

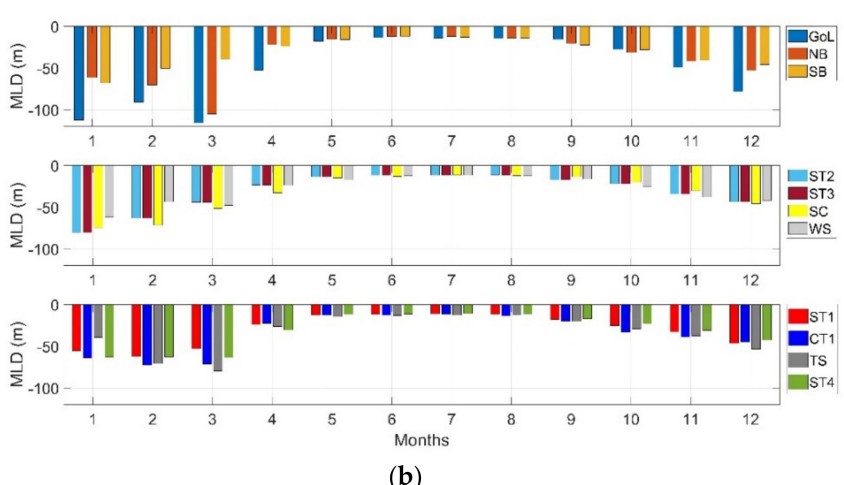

(**b**)

**Figure 4.** (**a**) Schematic vertical temperature profile highlighting the seasonal change of the three main oceanic layers; (**b**) the monthly climatological pattern of the MLD at the different sites, shown in the three areas that are discussed in the paper (northwestern Mediterranean, upper panel; Sardinia and Sicily Channel, central panel; and Tyrrhenian Sea, lower panel).

The study area can be divided into three main sectors (see circled groups of stations in Figure 1) when comparing their oceanographic features and MLDs: the sites in the Gulf of Lion and the Balearic Sea (GoL, NB, and SB), the sites enclosed in the Tyrrhenian Sea (ST1, CT1, TS, and ST4), and the sites encompassing the Sardinian and the Sicilian Channel (ST2, ST3, SC, and WS). The maximum values of the MLD (>100 m) are observed in January–March in the Gulf of Lion and the Northern Balearic Sea (Figure 4b), where, due to strong north-westerly winds, dense water formation events occur [112,113]. The fetch of these winds is also known to reach the Sicily Channel (where the climatological depth of the ML ranges from 60 to 80 m in winter; see Figure 4b). The present-day climatological depths of the Tyrrhenian ML during winter range from 40 m to 60 m (Figure 4b).

Here, in these sectors, the high percentages of l.c. *G. truncatulinoides* are recorded.

Based on this, we speculate that the increase in l.c. *G. truncatulinoides* highlighted in the Middle–Late Holocene transition is correlated to the re-establishment of the ML or to its deepening, at least, on a seasonal scale. In particular, l.c. *G. truncatulinoides* would have found a favorable habitat between 5 and 4 ka BP, as the stratification conditions, which characterized the deposition of Sapropel S1, had ended and the climate moved towards colder conditions, culminating with the 4.2 ka event.

These observations compliment other recent research. Perşoiu et al. (2019) suggest that the atmospheric blocking event induced by the strengthened Siberian High played an important role during the 4.2 ka BP event. Cold winters in Europe are associated with either blocking conditions over central Europe or the westward expansion of the high-pressure cell—the Siberian High—centered over Asia [114–117]. We suggest that the blocking conditions that occurred during the 4.2 ka event and during the MM triggered favorable oceanographic conditions for l.c. *G. truncatulinoides*.

A similar scenario was also proposed in order to explain the increase in l.c. *G. truncatulinoides* during the Little Ice Age (LIA) [30].

Furthermore, in the northwestern Mediterranean Sea, the percentage decrease in l.c. *G. truncatulinoides* during the second half of the 20th century was related to the reduced vertical mixing and lower surface productivity [118].

Therefore, we assume that l.c. *G. truncatulinoides* can be considered an excellent indicator of the presence of a deep ML in paleoceanographic reconstructions in the central and western Mediterranean Sea. In fact, prior to this time, the conditions were not adequate to allow a breeding population of *G. truncatulinoides* to become established, due to the mixed layer conditions that were ecologically prohibitive due to the complex depth migrations that this species undergoes.

## 7. Conclusions

The distributional patterns of l.c. *G. truncatulinoides* were analyzed in 10 cores collected from the central and western Mediterranean Sea (Gulf of Lion, North and South Balearic Sea, Tyrrhenian Sea, and Sicily Channel) over the last 6 ky BP. In all investigated sites, l.c. *G. truncatulinoides* shows a significant increase in abundance starting at c.a. 5–4 ky BP, approximating the base of the Meghalayan age (4.2 ka BP).

The significant increase in the abundance of l.c. *G. truncatulinoides* coincides with the end of African Humid Period when the Mediterranean Sea experienced a transition from a more humid climate to the present-day semi-arid climate. This climatic transition is likely responsible for the induction of winter intense deep vertical mixing and the subsequent enhancement of productivity in the mixed layer, favoring the proliferation of l.c. *G. truncatulinoides*. As a consequence, it is proposed that l.c. *G. truncatulinoides* can be used to identify variations in the depth of the ML and, therefore, the thermocline in paleoceanographic reconstructions in the Mediterranean Sea.

L.c. *G. truncatulinoides* can be also considered a potential tool for the chronological subdivision of the Middle–Late Holocene time interval and to approximate the base of the Late Holocene in Mediterranean marine records, linking the micropaleontological signal to the onset of paleoenvironmental conditions that are still active today in the central and western Mediterranean Sea.

**Author Contributions:** Conceptualization, G.M., F.L. and L.C.; methodology, G.M.; investigation, G.M.; resources, F.L., A.C.-H., A.C., T.B., I.C. and F.J.S.; data curation, G.M. and K.S.; writing—original draft preparation, G.M., L.C. and K.S.; writing—review and editing, G.M and K.S.; supervision, G.M, L.C. and F.L. All authors have read and agreed to the published version of the manuscript.

**Funding:** This research was financially supported by ERC-Consolidator TIMED project (REP-683237).

**Data Availability Statement:** The quantitative data of *G. truncatulinoides* are available on Pangaea data publisher.

**Acknowledgments:** Isabel Cacho acknowledges support from the ICREA Academia programe from the Generalitat de Catalunya.

**Conflicts of Interest:** The authors declare no conflict of interest.

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
