# Peer review of "Globorotalia truncatulinoides in the Mediterranean Basin during the Middle–Late Holocene: Bio-Chronological and Oceanographic Indicator"

_geosciences, doi:10.3390/geosciences12060244_

Round 1

Reviewer 1 Report

Revision of:

Globorotalia truncatulinoides in the Mediterranean basin during the Middle-Late Holocene: bio-chronological and oceanographic indicator

By Margaritelli and coauthors.

Margaritelli et al. compare abundance data for the planktonic foraminiferal species Globorotalia truncatulinoides from the western Mediterranean region, presenting evidence for an increase in abundance of the left-coiled morphotype around 4.5 kyr ago. They correlate such abundance increase across multiple sites and propose that it could serve as a bioevent to mark the transition from the Middle to the Late Holocene Stages in the Mediterranean. Further, the authors suggest that the increase in G. truncatulinoides could mark the onset of different environmental conditions in the western Mediterranean region, characterized by a drier and colder climate and a more pronounced seasonality, following the end of the African Humid Period.

I like this work by Margaritelli et al. as it provides a good preliminary assessment of the potential of a new planktonic foraminiferal bioevent to biostratigraphycally divide the late Middle to Late Holocene in the marine realm (previously only identified with geochemical proxies). If robustly dated and correlated across multiple west Mediterranean locations, the interval with peak abundance of G. truncatulinoides has potential to become a biohorizon for the region, because it occurs above a well-known sapropelic interval where the species is absent. The interpretation of peaks in abundance of G. truncatulinoides as paleoindicator for a deep mixed layer is also interesting and worth further investigation. I recommend publication after minor revisions to improve the clarity of the manuscript:

Main Suggestions:

1)    The distinction between data original of this study and data from literature is unclear. Please clearly state in the methods from which of the 11 study sites the authors provide new data and from which sites the data are from literature.

2)    For the sites where new data were generated, the authors should provide more details in the methods eg., age model? Sample resolution? Forams preservation?

3)    Given different sites have different age models, the age model uncertainty for each site should be provided.

4)   Figure 3 presents the main result of the paper but it is difficult to read. Please link each plot to its own X axis. Also, it is not clear what the gray bar is meant to highlight? I suggest to use different colors in the figure to highlight the wet and arid period described in the discussion.

5)    In paragraph 6.1, I would stick to the biostratigraphic interpretation of the G. truncatulinoides record and leave any ecological/environmental consideration for paragraph 6.2. In paragraph 6.2, the argument for G truncatulinoides as paleoindicator is a bit lost amongst lots of oceanographic details. I would use this paragraph to focus the discussion on G. truncatulinoides ecology and paleoecology (see for instance the works of Pallacks et al., 2021 and Boscolo-Galazzo et al., 2022) to set the context to argue for its use as deep mixed layer paleoindicator.

6)   Throughout the text please be consistent with using the coiling direction specification for G. truncatulinoides. Either always specify left/right coiled or never, as it becomes confusing.

7) Globortalia truncatulinoides has been recently renamed Truncorotalia truncatulinoides in the phylogenetic review by Aze et al. (2011). Consider following the updated nomenclature.

Minor Comments:

Please see annotated pdf for minor comments.

Author Response

Main Suggestions:

  • The distinction between data original of this study and data from literature is unclear. Please clearly state in the methods from which of the 11 study sites the authors provide new data and from which sites the data are from literature.

A new part has been inserted in the methods section that better specifies what is required. In any case, Table 1 shows all the information requested.

 2)    For the sites where new data were generated, the authors should provide more details in the methods eg., age model? Sample resolution? Forams preservation?

 A new part has been inserted in the methods section that better specifies what is required. In any case, Table 1 shows all the information requested.

  • Given different sites have different age models, the age model uncertainty for each site should be provided.

As regards the uncertainties of the various age models in Tab 1 and also in the text (methods section), please refer to the various papers in which this information is present.

 4)   Figure 3 presents the main result of the paper but it is difficult to read. Please link each plot to its own X axis. Also, it is not clear what the gray bar is meant to highlight? I suggest to use different colors in the figure to highlight the wet and arid period described in the discussion.

 Ok, done!

5)    In paragraph 6.1, I would stick to the biostratigraphic interpretation of the G. truncatulinoides record and leave any ecological/environmental consideration for paragraph 6.2. In paragraph 6.2, the argument for G truncatulinoides as paleoindicator is a bit lost amongst lots of oceanographic details. I would use this paragraph to focus the discussion on G. truncatulinoides ecology and paleoecology (see for instance the works of Pallacks et al., 2021 and Boscolo-Galazzo et al., 2022) to set the context to argue for its use as deep mixed layer paleoindicator.

Thank you for your suggestion. We have tried, as far as possible, to modify the text according to the instructions of the reviewer.

6)   Throughout the text please be consistent with using the coiling direction specification for G. truncatulinoides. Either always specify left/right coiled or never, as it becomes confusing.

 Done. Whenever necessary, the type of coling was specified.

7) Globortalia truncatulinoides has been recently renamed Truncorotalia truncatulinoides in the phylogenetic review by Aze et al. (2011). Consider following the updated nomenclature.

      Done. This information and its reference have been reported in the text.

Minor Comments:

Please see annotated pdf for minor comments. All the minor comments reported in the pdf have been integrated. The request to make the data public was accepted. Once the paper has been published, as also reported in the Acknowledgment, the procedure for publishing the data on Pangaea will be started.

Reviewer 2 Report

Dear Authors,

Thank you for submitting the manuscript for your study, it was a pleasure to read and I look forward to it being published.

I have only very minor concerns with the manuscript which you will find attached as comments to the PDF file I reviewed.

As you will see my primary comments and corrections concern very minor reference edits, and spelling and grammar errors. The science of the study is good, and well presented, however there are a few sentences in the final paragraphs which I believe need to be slightly changed to better fit the results of your study.

I look forward to your manuscript being published, and referencing it in the future.

All the best to the Authors.

Author Response

Dear Reviewer and Editor,

all the minor comments reported in the pdf have been integrated. The lack of italic character in the name of the species G. truncatulinoides is not attributable to our lack, I believe it is a problem of pagination of the generated pdf. The same for the position of the captions of figs. 1 and 2 and due to the poor readability of fig. 4b, already noted by the first reviewer.

Best,

Giulia Margaritelli

Reviewer 3 Report

The paper " Globorotalia truncatulinoides in the Mediterranean basin during the
Middle-Late Holocene: bio-chronological and oceanographic indicator" by Margaritelli et al., is well-documented and well structured. It has clear objectives, and methods are adequately described.

Interpretations and conclusions are supported by constructive and well-designed arguments. Although it is a regional application, oceanographic results could be a reference for possible analogue outside the Mediterranean, whereas the biostratigraphic results are greatly useful for core stratigraphy in the Mediterranean.

I see no objection against publishing it in the submitted form.

Minor observations:

Please use "italic" font for "G. truncatulinoides " in title and abstract

 Line 121 - caption of Fig. 2

Line 125 - caption of Fig. 1

Line 130 - last line of  tab. 1: from "Trias et al., 2021" to: "Trias-Navarro et al., 2021"

Line 147 - avoid abbreviation in chapter title and use italic font: "Globorotalia truncatulinoides:..."

Line 193 - delete "recent"

 line 332 - delete "-"

 Figure 4a could be improved, texts are not clear, mainly in the upper part.

Author Response

Minor observations:

Please use "italic" font for "G. truncatulinoides " in title and abstract

In our original file this error is not reported probably is a pagination error of the generated pdf.

 Line 121 - caption of Fig. 2

In our original file this error is not reported probably is a pagination error of the generated pdf.

Line 125 - caption of Fig. 1

In our original file this error is not reported probably is a pagination error of the generated pdf.

Line 130 - last line of  tab. 1: from "Trias et al., 2021" to: "Trias-Navarro et al., 2021"

Ok, DONE

Line 147 - avoid abbreviation in chapter title and use italic font: "Globorotalia truncatulinoides:..."

Ok, DONE

Line 193 - delete "recent"

ok, DONE

 line 332 - delete "-"

ok DONE

 Figure 4a could be improved, texts are not clear, mainly in the upper part.

In the pdf generated by the journal the background of this figure has become black and therefore unclear. We attach to the re-submission a new file of fig. 4b to be inserted with a white background.

Best,

Giulia Margaritelli